# GABA Regulates Phenolics Accumulation in Soybean Sprouts under NaCl Stress

**DOI:** 10.3390/antiox10060990

**Published:** 2021-06-21

**Authors:** Yunyun Zhao, Chong Xie, Pei Wang, Zhenxin Gu, Runqiang Yang

**Affiliations:** College of Food Science and Technology, Nanjing Agricultural University, Nanjing 210095, China; 2020108055@stu.njau.edu.cn (Y.Z.); xiechong@njau.edu.cn (C.X.); wangpei@njau.edu.cn (P.W.); guzx@njau.edu.cn (Z.G.)

**Keywords:** NaCl stress, GABA, phenolics components, accumulation

## Abstract

NaCl stress causes oxidative stress in plants; γ-aminobutyric acid (GABA) could alleviate such abiotic stress by enhancing the synthesis of phenolics, but the underlying mechanism is not clear. We investigated the effects of GABA on phenolics accumulation in soybean sprouts under NaCl stress by measuring changes in the content of physiological biochemicals and phenolic substances, in the activity and gene expression of key enzymes, and in antioxidant capacity. GABA reduced the oxidative damage in soybean sprouts caused by NaCl stress and enhanced the content of total phenolics, phenolic acids, and isoflavones by 16.58%, 22.47%, and 3.75%, respectively. It also increased the activities and expression of phenylalanine ammonia lyase, cinnamic acid 4-hydroxylase, and 4-coumarate coenzyme A ligase. Furthermore, GABA increased the activity of antioxidant enzymes and the antioxidant capacity. These events were inhibited by 3-mercaptopropionate (an inhibitor for GABA synthesis), indicating that GABA mediated phenolics accumulation and antioxidant system enhancement in soybean sprouts under NaCl stress.

## 1. Introduction

Soybean (*Glycine max* L.) is a key source of protein, fat, and functional carbohydrates [1]. In addition, it contains some special phytochemicals [2] and is popular in East Asian countries, such as China, South Korea, and Japan. Soybean sprouts are broadly consumed in China. During germination, endogenous enzymes are activated and degrade stored nutrients in the seeds [3,4]. This also changes the biochemical, nutritional, and sensory properties and induces the synthesis of a series of bioactive substances including phenolic acids and isoflavones [5]. Therefore, germination is a convenient and feasible way to fully utilize soybean resources and improve their nutritional quality [6]. Soybean sprouts may accumulate high level of phenolic compounds to protect themselves from the oxidative damage caused by high levels of reactive oxygen species (ROS) under salt stress [7].

γ-aminobutyric acid (GABA) is a four-carbon non-protein amino acid with a variety of physiological functions [8]. In recent years, its physiological functions as a signal molecule in reducing abiotic stress in plants have been deeply studied [9,10]. Past research has shown that GABA itself exhibits ROS scavenging ability [11]. Treatment of caragana roots with GABA under salt stress induced a variety of physiological and biochemical responses, such as signal transmission, protein degradation regulation, hormone synthesis, ROS formation, and polyamine metabolism [12], which indicates that GABA might be involved in the expression and regulation of salt-related genes in plants under salt stress. The content of GABA and phenolics in plants increased under salt stress [13] and exogenous GABA treatment [9]. Since phenolics are not related to GABA metabolism, it is speculated that GABA acts as a signal molecule to mediate the synthesis of phenolics in soybean sprouts. Whether GABA acts as a signal molecule and the exact mechanism of its action are uncertain. Therefore, it is necessary to explore the mechanism by which GABA regulates enzyme activity and accumulation of phenolics under adverse conditions.

In this study, the effects of GABA and its inhibitor 3-mercaptopropionate (3-MP) on the content of phenolics and the activity of key enzymes for phenolics synthesis including phenylalanine ammonia lyase (PAL), cinnamate-4-hydroxylase (C4H), and 4-coumarate coenzyme A ligase (4CL) were studied to reveal the regulating effect of GABA on the synthesis of phenolics in soybean sprouts under NaCl stress.

## 2. Materials and Methods

### 2.1. Materials and Reagents

Soybean seeds (Dongsheng No. 1), harvested in 2019 in Heilongjiang province, China, were stored at −20 °C before use. Methanol (AR), ethyl acetate, 2,2’-diphenyl-1-picrylhydrazyl (DPPH), 2,2’-azino-bis (3-ethylbenzothiazoline-6-sulfonic acid) diammonium salt (ABTS), nicotinamide adenine dinucleotide phosphate (NADPH), cinnamic acid were purchased from Sinopharm Chemical Reagent Co., Ltd. (Shanghai, China). Mercaptoethanol, coenzyme A (CoA), trolox, acetonitrile, methanol (HPLC grade) were purchased from Maclean Biotechnology Co., Ltd. (Shanghai, China). The standards of phenolic acids and isoflavone were purchased from Sigma-Aldrich Chemical Co. (Shanghai, China).

### 2.2. Experimental Design

After removal of impurities and washing, 70 g soybean seeds was sterilized with a 0.5% (*w*/*v*) sodium hypochlorite solution for 15 min, washed to neutral pH with distilled water, and then soaked in distilled water at 30 °C for 6 h. The soaked seeds were placed on the seed tray of a germination machine and germinated in an incubator (30 °C), where they were sprayed once every hour for 1 min with a designed culture solution, as follows: a. Control (CK): the soybean seeds were cultured with distilled water. b. NaCl treatment (N): the soybean seeds were sprayed with 40 mM NaCl. c. NaCl + GABA treatment (NG): the seeds were sprayed with 40 mM NaCl and 5 mM GABA. d. NaCl + 3-MP treatment (NM): the seeds were sprayed with 40 mM NaCl and 0.2 mM 3-MP. e. NaCl + 3-MP treatment + GABA (NMG): the seeds were sprayed with 40 mM NaCl, 0.2 mM 3-MP, and 5 mM GABA. The culture solutions were changed every 24 h.

### 2.3. Sprout Length, Fresh Weight, and Dry Weight Determination

Sprout length was measured with vernier calipers, in mm. Fresh weight and dry weight were determined by weighing a certain amount of fresh soybean sprouts accurately (fresh weight) and then weighing them after freeze-drying in a vacuum freeze dryer for 48 h (dry weight).

### 2.4. Phenolics Content Assay

#### 2.4.1. Extraction of Free Phenolics 

Free phenolics were extracted according to Chen et al. [14] with slight modifications. The freeze-dried soybean sprouts were pulverized and screened through a 60-mesh sieve. Two grams of powder was degreased with hexane for 3 times and shaken with 80% methanol (*v*/*v*) for 1 h (25 °C, 200 rpm) under N_2_ in the dark at 25 °C. After shaking, the sample was centrifuged (10,000× *g*, 4 °C) for 15 min (extraction was repeated for 3 times). The supernatants were filtered, combined into a flat bottom flask, and rotated to dryness at 40 °C. The dried extract was brought to 10 mL with 50% methanol (*v*/*v*) as the free phenolics extract, which was then filled with N_2_ and stored at −20 °C for analysis.

#### 2.4.2. Extraction of Bound Phenolics

Bound phenolic compounds were extracted according to Chen et al. [13]. The residue from free phenolics extraction was hydrolyzed with NaOH (2 mol/L) in a shaker in the dark for 4 h (25 °C, 200 rpm). The hydrolysate was adjusted to a pH ranging from 1.5 to 2.0 with HCl (6 M). After 15 minutes of ethyl acetate extraction, the mixture was centrifuged (10,000× *g*, 4 °C) for 5 min. The ethyl acetate layers from 3 centrifugations were combined in a flat-bottomed flask and rotated to dryness at 40 °C. The dried extract was brought to 10 mL with 50% methanol (*v*/*v*) as the bound phenolics extract, which was then filled with N_2_ and stored at −20 °C for analysis.

#### 2.4.3. Determination of Total Phenolic Content

The total phenolic content (TPC) was measured according to the Folin’s phenol method [15] with slight modifications. Ten-fold diluted Folin’s phenol reagent was added to 200 μL of the above-mentioned diluted phenolics extract and vortexed. After 5 min from vortexing, a Na_2_CO_3_ solution (75 g/L) was added. After mixing, the mixture was maintained in the dark for 2 h at 25 °C. Then, the absorbance at 765 nm was measured using a 50% methanol solution as the blank control. A standard curve was prepared using gallic acid as a standard (0–90 μg/mL). The TPC was expressed as mg Gallic Acid Equivalent (GAE)/100 g Dry weight (DW).

#### 2.4.4. Determination of Free and Bound Phenolic Acid Content

The free and bound phenolics extract were filtered through an organic filter (0.45 μm, Tianjin Branch billion Lung Experimental Equipment Co.,Ltd., Tianjin, China) and analyzed by high-performance liquid chromatography (HPLC) (Shimadzu LC-20A, column C18 110A, 5 μm particle size, 4.6 × 150 mm, Phenomenex, Torrance, CA, USA) [13]. The mobile phase A was the 0.1% acetic acid solution, and the mobile phase B was methanol containing 0.1% of acetic acid. The flow rate was 0.9 mL/min, and the running time was 75 min. The column temperature was 35 °C, and the measurement wavelength was 280 nm. The phenolic acid content was calculated according to a standard curve (0–90 μg/mL), and the results were expressed in μg/g DW.

#### 2.4.5. Determination of Isoflavones Content

The extraction of isoflavones was conducted according to Jiao et al. [16]. The extract was filtered through a 0.45 µm filter membrane, and 20 μL of the filtered isoflavone extract was analyzed by HPLC (Agilent 1200, Agilent Technologies Co. Ltd., Shanghai, China) with a Zorbax SB-C18 column (5 μm particle size, 4.6 × 150 mm, Agilent Technologies Inc., CA, USA) and a Variable Wavelength Detector(VWD) detector. The mobile phase A was 0.1% acetic acid solution, and the mobile phase B was acetonitrile containing 0.1% acetic acid. The flow rate was 1.0 mL/min, and the running time was 52 min. The column temperature was 35 °C, the detection wavelength was 260 nm. The isoflavones content was calculated according to a standard curve (0–65 μg/mL), and the results were expressed as μg/g DW.

### 2.5. Determination of Key Enzymes Activity for Phenolics Synthesis

PAL activity was determined as described by Assis et al. [17]. After zeroing with distilled water, the absorbance was measured at 290 nm. A unit of PAL activity (U) was defined as an increase in absorbance of the enzymatic reaction system per gram of soybean sprouts (fresh weight) per minute by 0.01, which was expressed as 0.01△OD_290_/h•g Fresh weight (FW).

C4H activity measurement was conducted according to the method of Lamb and Rubery [18]. The absorbance was measured at 340 nm. A unit of C4H activity (U) was defined as an increase in absorbance of the enzymatic reaction system per gram of soybean sprouts (fresh weight) per minute by 0.01, which was expressed as 0.01△OD_340_/min•g FW.

For 4CL activity measurement, the method of Han et al. [19] was used directly. After zeroing with distilled water, the absorbance was measured at 333 nm. A unit of 4CL activity (U) was shown as an increase in absorbance of the enzymatic reaction system per gram of soybean sprout (fresh weight) per minute by 0.01, which was expressed as 0.01△OD_333_/min•g FW.

### 2.6. Gene Expression

Total RNA was isolated according to Ma et al. [9]. Frozen soybean sprouts were ground in liquid nitrogen to fine powders, and total RNA was extracted using a MiniBEST PlantRNA Extraction kit (TaKaRa, Shiga, Japan). Real-time PCR analysis was conducted using SYBR *Premix Ex Taq* kit (TaKaRa, Shiga, Japan). Three biological replicates were performed for quantitative assays for each gene. The primers used in the present study are shown in Table 1 with *EF1b* as the internal reference. The reaction procedure was: 95 °C, 30 s, and then for 40 cycles: 95 °C, 5 s; 60 °C, 34 s; ended at 95 °C, 15 s; 60 °C, 1 min; 95 °C, 15 s; cooled to 4 °C finally.

### 2.7. Antioxidant Capacity Determination

#### 2.7.1. DPPH Free-Radical Scavenging Ability

According to Chen et al. [13], a 50% methanol solution was used as a blank control, and a standard curve was prepared with Trolox. The DPPH free-radical scavenging capacity was calculated as follows:$$DPPH\;free-radical\;scavenging\;ability\;\left(\%\right)=\left(1-\frac{A_{s}-A_{j}}{A_{control}}\right)\times 100$$

*A_s_*, *A_j_*, and *A_control_* are the OD_515_ values of the sample groups, pure methanol group, and control group, respectively. The results were expressed in μmol TE/g DW.

#### 2.7.2. ABTS Free-Radical Scavenging Activity

We referred to the method of Chen et al. [13]: a 50% methanol solution was used as a blank control, and a standard curve was prepared with Trolox. The ABTS free-radical scavenging capacity was calculated as follows:$$ABTS\;free\;radical\;scavenging\;ability\;\left(\%\right)=\left(1-\frac{A_{s}-A_{j}}{A_{control}}\right)\times 100$$

*A_s_*, *A_j_*, and *A_control_* are the OD_734_ values of the sample groups, pure methanol group, and control group, respectively. The results were expressed in μmol TE/g DW.

### 2.8. Antioxidant Enzyme Activity

Peroxidase (POD) activity was determined using the method of Chisari et al. [20]. Catalase (CAT) activity was determined by the method of Wang et al. [21]. Superoxide dismutase (SOD) activity was measured using a kit (A001, Nanjing Jiancheng Biotechnology Research Institute, Nanjing, China), following the instruction manual.

### 2.9. Data Processing and Statistical Analysis

The experiments were set up with 3 biological and technical repetitions, and the results are presented as mean ± standard deviation (SD). The Ducan’s multiple range test was used for the significance (at the 0.05 level) test by the software GraphPad Prism 7.0 (GraphPad Software Inc., San Diego, CA, USA).

## 3. Results

### 3.1. Effects of GABA on Length, Fresh Weight, and Dry Weight of Sprouts under NaCl Stress

The NaCl and NaCl plus 3-MP treatments inhibited the growth of soybean sprouts in length by 12.50% and 19.45%, respectively, compared with the control (Figure 1A,B). Adding GABA alleviated the inhibition of NaCl stress by 24.01% and that of 3-MP treatment by 34.70%. Both treatments had the greatest effect on hypocotyl growth. Therefore, the growth of soybean sprouts under NaCl stress appeared to be regulated by GABA, which could relieve NaCl stress and promote the growth of the soybean sprouts. Under NaCl treatment, the fresh weight of the soybean sprouts after GABA treatment significantly increased by 13.37%, while there was no significant change under other treatments (Figure 1C). The 3-MP treatment significantly reduced the dry weight of the soybean sprouts under NaCl stress.

The effect of GABA and its inhibitor on the TPC of soybean sprouts under NaCl stress is shown in Figure 1D. The TPC increased significantly by 19.49% under NaCl treatment and 38.38% under NaCl plus GABA treatment, compared with CK. Under NaCl stress, GABA increased TPC by 16.58%, but 3-MP inhibited phenolics synthesis by 2.77%. Application of GABA with 3-MP could relieve the inhibition, increasing the TPC by 6.82% compared with the 3-MP treatment. The content of free phenolics in the soybean sprouts was about 20 times higher than that of bound phenolics, and NaCl and GABA mainly induced the synthesis of bound phenolics. The content of free phenolics and bound phenolics in the soybean sprouts under NaCl treatment increased by 3.61% and 18.68%, respectively, compared with CK. The content of the free and bound forms of phenolics increased by 15.30% and 44.37% after GABA treatment under NaCl stress. The content of bound phenolics decreased by 15.78% after 3-MP treatment, but further GABA treatment increased these values by 5.60% and 37.49%, respectively, compared with the 3-MP treatment.

### 3.2. Effects of GABA on Phenolic Acid and Isoflavone Content in Soybean Sprouts under NaCl Stress

The effects of GABA and its inhibitor on the content of phenolic acids in soybean sprouts under NaCl stress are shown in Table 2. The phenolic acids mainly exist in free form in soybean sprouts. A total of six phenolic acids were detected. The main types of phenolic acids were syringic acid, *p*-coumaric acid, and ferulic acid, which accounted for about 85% of the content of the total phenolic acids. The highest content of total phenolic acids was found after treatment with NaCl plus GABA. The content of *p*-coumaric acid was the highest among the free phenolic acids, while that of syringic acid was the highest among the bound phenolic acids.

The NaCl and NaCl plus GABA treatments increased the total phenolic acids content by 59.75% and 30.43%, respectively, compared with CK. The treatment with 3-MP under NaCl stress reduced the content of total phenolic acids by 36.24%, while the addition of GABA increased it by 48.46%. Among the main phenolic acids, NaCl treatment increased the content of syringic acid, *p*-coumaric acid, and ferulic acid by 20.60%, 29.68%, and 34.71%, respectively, compared with CK, and GABA increased these contents by 27.92%, 19.99%, and 6.71%. However, the content decreased significantly after the addition of 3-MP under NaCl stress, and re-adding GABA significantly alleviated the decrement.

We found that the NaCl and NaCl + GABA treatments increased the isoflavone content by 18.58% and 23.03%, respectively, compared with CK (Table 3). However, the 3-MP treatment significantly decreased the isoflavone content under NaCl stress, while further addition of GABA reduced this effect. The content of malonyldaidzin was the highest in CK, followed by that of malonylgenistin. In contrast, under NaCl treatment, the content of malonylgenistin became the highest, followed by that of malonyldaidzin. Compared with CK, a new isoflavone, genistein, was detectable in other treatment groups. Under NaCl stress, the content of daidzin and malonylglycitin in soybean sprouts was significantly reduced by 10.51% and 16.92%, respectively. The content of daidzein and malonylgenistin increased by 58.89% and 71.17%, respectively, compared with CK. Under NaCl + GABA treatment, the content of malonylgenistin increased by 7.11%, while the content of daidzein and genistein decreased by 25.58% and 6.89%, respectively, compared with NaCl treatment. On the basis of NaCl stress, 3-MP significantly reduced the isoflavone content, except for daidzin, glycitin, and malonylgenistin, but the addition of GABA increased the content of these molecules, except for malonylgenistin.

### 3.3. Effects of GABA on Key Enzymes Activity and mRNA Levels in Soybean Sprouts under NaCl Stress

The effects of GABA and its inhibitor on the activities of PAL, C4H, and 4CL under NaCl stress are shown in Figure 2. After NaCl treatment, PAL, C4H, and 4CL activities increased by 40.28%, 20.49%, and 194.03%, respectively, compared with CK. Based on NaCl treatment, GABA further increased the activity of PAL, C4H, and 4CL by 41.82%, 23.16%, and 73.86%, respectively, while 3-MP decreased their activity by 22.65%, 45.07%, and 77.41%, respectively. The further addition of GABA increased the enzymes’ activities by 65.37%, 18.44%, and 254.51%, compared with the NaCl + 3-MP treatment. The trends of the relative expression levels of *PAL*, *C4H*, and *4CL* appeared similar to those of their activity (Figure 2B,D,F). The expression of *PAL* and *4CL* was more sensitive to GABA treatment under NaCl stress.

### 3.4. Effects of GABA on Free-Radical Scavenging Ability and Antioxidant Enzyme Activity in Soybean Sprouts under NaCl Stress

The effects of GABA and its inhibitor on the antioxidative system of soybean sprouts under NaCl stress were measured (Figure 3A,B). Compared with CK, treatments with NaCl, NaCl + GABA, NaCl + 3-MP, and NaCl + 3-MP + GABA increased ABTS and DPPH free-radical scavenging ability, up to a relatively high level upon treatments including GABA. Compared with the bound extract, the free extract displayed a higher free-radical scavenging ability, which was 3–20 times higher than that of the bound extract. As shown in Figure 3C,D, similar to what observed for the free-radical scavenging ability, the activities of CAT, POD, and SOD in all treated groups were significantly higher than those in the control (CK). In addition, GABA treatment could enhance the antioxidant enzyme activity of soybean sprouts. GABA had a greater effect on the increment in the activity of these antioxidant enzymes, indicating that GABA could enhance the antioxidant systems and relieve the oxidative damage in soybean sprouts caused by NaCl stress.

## 4. Discussion

The present study showed that NaCl stress increased the content of total phenolics, phenolic acids, and isoflavones in soybean sprouts and their antioxidant capacity [7]. GABA induced a further increase in total phenolics content in soybean sprouts upon NaCl stress (Figure 1D). In order to further investigate whether GABA was involved in the synthesis of phenolics, a GABA synthesis inhibitor (3-MP) was added under NaCl stress. We found that under NaCl stress, the content of total phenolics decreased in the presence of 3-MP, while it increased in the presence of 3-MP plus GABA, compared with that under 3-MP treatment. Therefore, GABA alleviated the inhibition of 3-MP on soybean phenolics synthesis and participated in the synthesis of phenolics under NaCl stress.

In this study, the changes in the contents of phenolic acid and isoflavones in soybean sprouts under each treatment were positively correlated with the level of total phenolics, but the distribution of each phenolic acid and isoflavone was extremely uneven (Figure 2 and Figure 3). The content of free phenolic acids was significantly higher than that of the bound form, which might be due to the partial conversion of bound phenolic compounds to the free form after soybean seeds germination [13]. NaCl stress increased the content of isoflavone in soybean sprouts because NaCl stress activated the phenylpropane pathway, leading to isoflavone enrichment [22]. Under NaCl stress, the content of malonylgenistin in soybean sprouts increased significantly by 58.89% after GABA treatment, but the content of free isoflavone (daidzein) decreased. These results indicated that GABA treatment promoted the acetylation and malonylation of isoflavone, which plays a crucial role in the response of soybean sprouts to NaCl stress [23,24]. After 3-MP treatment under NaCl stress, the content of phenolic acid and isoflavone in soybean sprouts decreased, while the addition of GABA could partially reverse this reduction, indicating that GABA played a greater role in promoting phenolics synthesis.

This study showed that GABA treatment significantly enhanced PAL, C4H, and 4CL activities in soybean sprouts under NaCl stress. However, the activities of these enzymes were significantly inhibited by 3-MP and restored after re-addition of GABA, indicating that GABA was involved in the regulation of the activity of key enzyme in the phenolics synthesis of soybean sprouts under NaCl stress. These results indicate that GABA acted as a signal molecular to mediate phenolics synthesis under NaCl stress. The addition of exogenous GABA can further promote the signal transduction of endogenous GABA [25], thereby increasing gene expression and activity in soybean sprouts of key enzymes (Figure 2) in phenolics synthesis and promoting the synthesis of phenolics. The 3-MP treatment demonstrated that GABA transmitted signals to positively regulate phenolics synthesis in soybean sprouts under NaCl stress. Hence, GABA metabolism was critical for phenolics synthesis in soybean sprouts, and the exogenous GABA further increased the phenolics levels via the enhancement of activity and gene expression of key enzymes in the phenylpropane pathway [26,27].

In addition, this study explored the effect of GABA metabolism on antioxidant systems of soybean sprouts under NaCl stress (Figure 3). The results showed that, similar to the change observed in total phenolics content, NaCl stress increased the activities of CAT, POD, and SOD in soybean sprouts and significantly enhanced DPPH and ABTS free-radical scavenging capacities. Gharibi et al. [28] reported that under moderate drought stress, celery had the highest TPC and antioxidant capacity. Therefore, the antioxidant capacity of soybean sprouts is positively correlated with the content of phenolics under NaCl stress. Rezaei et al. [29] reported that GABA could significantly increase the proline content and CAT activity to promote tolerance of black cumin to water deficit stress. In this study, under NaCl stress, GABA treatment enhanced the activity of antioxidant enzymes in soybean sprouts and improved the ABTS and DPPH free-radical scavenging abilities (Figure 3A,B), which is consistent with the change in the content of phenolics in soybean sprouts (Figure 1D). This indicates that GABA treatment could promote the accumulation of phenolics in soybean sprouts and improve their antioxidant capacity. Hence, GABA appears to regulate the activity of phenolic synthetase in soybean sprouts and promote phenolic enrichment, thereby enhancing antioxidant capacity and stress tolerance and strengthening the defense of soybean sprouts und NaCl stress.

## 5. Conclusions

Syringic acid, *p*-coumaric acid, and ferulic acid are the main identified phenolic acids, while alonyldaidzin and malonylgenistin are the main identified isoflavones in soybean sprouts. GABA is involved in the synthesis of phenolics in soybean sprouts under NaCl stress by regulating the activity and gene expression of key enzymes. In addition, GABA treatment promoted the acetylation and malonylation of isoflavone, which might be crucial for the increase of free-radical scavenging ability in soybean sprouts.

## Figures and Tables

**Figure 1 antioxidants-10-00990-f001:**
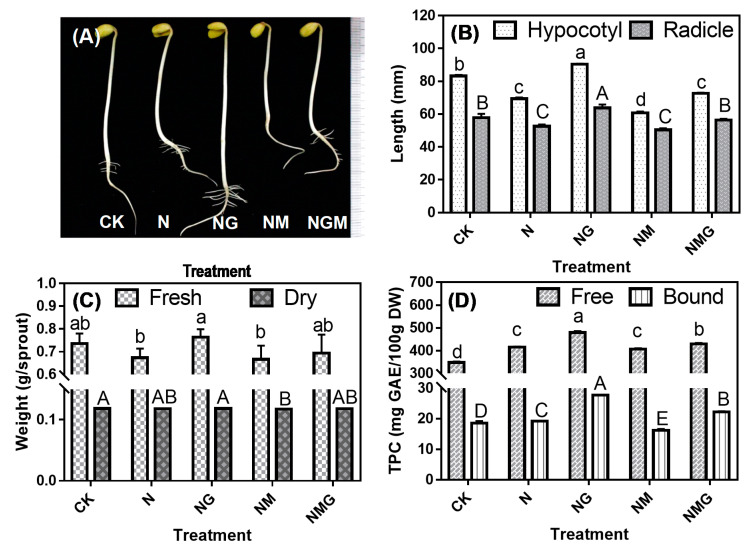
Effects of γ-aminobutyric acid (GABA) treatment on morphological status (**A**), length (**B**), weight (**C**), and phenolics content (**D**) of soybean sprouts under NaCl stress (*n* = 3). CK indicates distilled water treatment, N indicates 40 mM NaCl treatment, NG indicates 40 mM NaCl + 5 mM GABA treatment, NM indicates 40 mM NaCl + 0.2 mM 3-MP treatment, and NGM indicates 40 mM NaCl + 0.2 mM 3-MP + 5 mM GABA treatment. Error bars represent the standard deviation. Values in the same measured category bearing different uppercase letters (A–E) or lowercase letters (a–d) are significantly different (*p* < 0.05).

**Figure 2 antioxidants-10-00990-f002:**
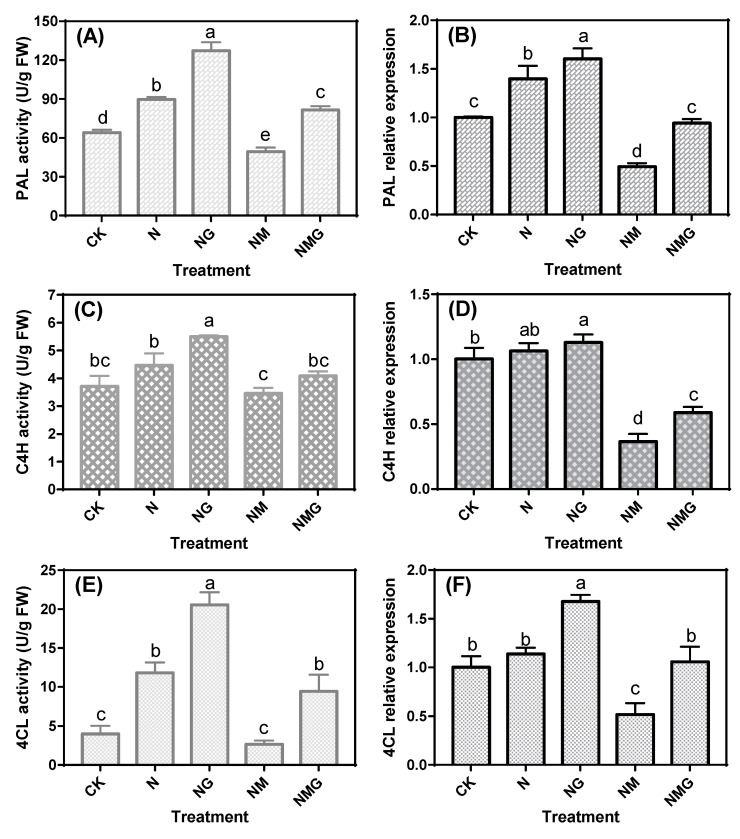
Effects of GABA treatment on phenylalanine ammonia lyase (PAL; **A**), cinnamic acid 4-hydroxylase (C4H; **C**), 4-coumarate coenzyme A ligase (4CL; **E**) activity and their mRNA levels (**B**,**D,F**) in soybean sprouts under NaCl stress (*n* = 3). CK, N, NG, NM, and NMG indicate distilled water treatment, 40 mM NaCl treatment, 40 mM NaCl + 5 mM GABA treatment, 40 mM NaCl + 0.2 mM 3-MP treatment, and 40 mM NaCl + 0.2 mM 3-MP + 5 mM GABA treatment, respectively. Error bars represent standard deviation. Values in the same measured category bearing different letters (a–e) are significantly different (*p* < 0.05).

**Figure 3 antioxidants-10-00990-f003:**
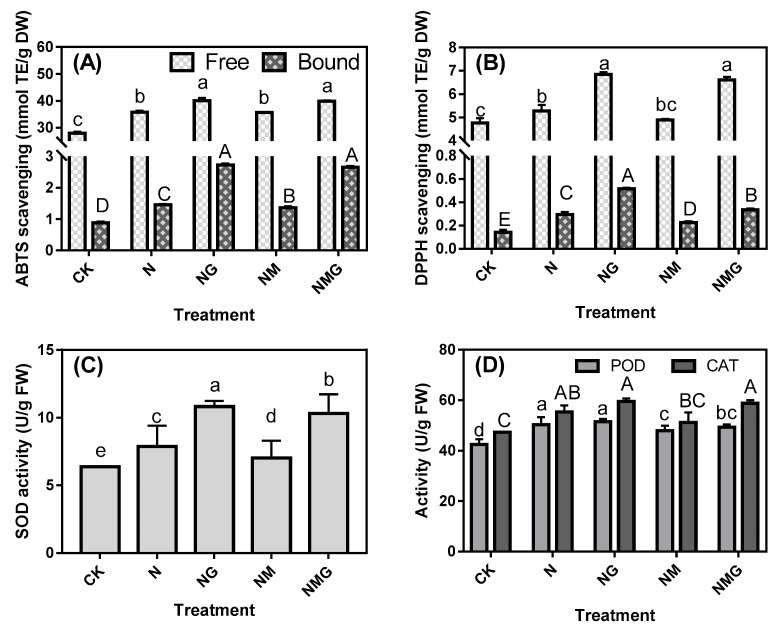
Effects of GABA treatment on 2,2’-azinobis (3-ethylbenzthiazoline-6-sulphonic acid) (ABTS; **A**) and1,1-Diphenyl-2-picrylhydrazyl radical 2,2-Diphenyl-1-(2,4,6-trinitrophenyl) hydrazyl (DPPH; **B**) Super oxide dimutese (SOD; **C**), CAT, and peroxidase (POD; **D**) activity in soybean sprouts under NaCl stress. CK, N, NG, NM, and NMG indicate distilled water treatment, 40 mM NaCl treatment, 40 mM NaCl + 5 mM GABA treatment, 40 mM NaCl + 0.2 mM 3-MP treatment, and 40 mM NaCl + 0.2 mM 3-MP + 5 mM GABA treatment, respectively. Error bars represent standard deviation. Values in the same measured category bearing different uppercase letters (**A**–**E**) or lowercase letters (a–e) are significantly different (*p* < 0.05).

**Table 1 antioxidants-10-00990-t001:** Primer sequences for genes related to phenolics synthesis in soybean.

Gene	Primer Name	Primer Sequences (5’→3’)
*PAL*	Sense	CTACCATCACCAATGGGAGCC
	Ant-sense	CTCCCCAGTTTAACGGATCACT
*C4H*	Sense	TGGCCTGCTAATGGGTATTGT
	Ant-sense	ACACAAATACTGGCTCTGCT
*4CL*	Sense	AGTCGCAGCCTTTCCATCAA
	Ant-sense	GACGATGTAGCGGATGTCGT
*EF1b*	Sense	CCACTGCTGAAGAAGATGATGATG
	Ant-sense	AAGGACAGAAGACTTGCCACTC

**Table 2 antioxidants-10-00990-t002:** Effects of GABA treatment on phenolic acids content in soybean sprouts under NaCl stress (*n* = 3).

Phenolic acid	Treatment	Content (µg/g DW)		
		Free	Bound	Total
*p*-hydroxybenzoic acid	CK	17.46 ± 0.13 ^d^	3.06 ± 0.28 ^b^	20.52 ± 0.21 ^d^
	N	28.23 ± 0.45 ^b^	3.41 ± 0.14 ^b^	31.64 ± 0.31 ^b^
	NG	37.15 ± 4.67 ^a^	4.67 ± 0.17 ^a^	41.82 ± 4.84 ^a^
	NM	21.31 ± 1.01 ^cd^	1.73 ± 0.33 ^d^	23.04 ± 1.34 ^c^
	NMG	26.43 ± 0.11 ^bc^	2.50 ± 0.27 ^c^	28.93 ± 0.16 ^b^
Vanillic acid	CK	70.74 ± 0.05 ^d^	55.59 ± 0.64 ^b^	126.33 ± 0.67 ^d^
	N	129.08 ± 0.15 ^b^	62.77 ± 2.18 ^a^	191.84 ± 2.03 ^ab^
	NG	149.21 ± 19.13 ^a^	66.03 ± 2.08 ^a^	215.24 ± 21.21 ^a^
	NM	90.32 ± 0.27 ^c^	23.28 ± 4.68 ^d^	113.60 ± 4.94 ^c^
	NMG	99.25 ± 0.93 ^c^	42.27 ± 0.17 ^c^	141.52 ± 1.11 ^b^
Syringic acid	CK	241.59 ± 3.31 ^c^	223.09 ± 7.93 ^c^	464.68 ± 11.33 ^d^
	N	265.03 ± 0.68 ^b^	295.37 ± 5.52 ^b^	560.40 ± 4.84 ^c^
	NG	379.63 ± 0.81 ^a^	337.20 ± 0.44 ^a^	716.84 ± 1.25 ^a^
	NM	199.06 ± 1.31 ^d^	106.79 ± 1.48 ^d^	305.85 ± 0.17 ^e^
	NMG	386.42 ± 10.73 ^a^	224.66 ± 5.35 ^c^	611.08 ± 10.38 ^b^
*p*-coumaric acid	CK	791.14 ± 2.04 ^c^	117.75 ± 1.14 ^c^	908.89 ± 2.27 ^c^
	N	1043.82 ± 19.17 ^b^	134.85 ± 4.80 ^b^	1178.68 ± 14.37 ^b^
	NG	1253.58 ± 4.12 ^a^	160.73 ± 0.14 ^a^	1414.31 ± 4.26 ^a^
	NM	726.63 ± 72.17 ^c^	83.77 ± 0.60 ^d^	810.40 ± 71.57 ^d^
	NMG	893.92 ± 16.75 ^bc^	169.08 ± 6.21 ^a^	1063 ± 16 ^bc^
Ferulic acid	CK	197.89 ± 4.21 ^b^	18.01 ± 1.13 ^c^	215.90 ± 4.54 ^b^
	N	267.82 ± 0.32 ^a^	23.01 ± 2.29 ^b^	290.83 ± 1.97 ^ab^
	NG	283.14 ± 0.90 ^a^	27.22 ± 1.04 ^a^	310.35 ± 0.14 ^a^
	NM	181.80 ± 17.39 ^b^	14.13 ± 1.15 ^d^	195.93 ± 18.54 ^c^
	NMG	271.67 ± 8.20 ^a^	20.44 ± 0.40 ^bc^	292.10 ± 8.60 ^ab^
Erucic acid	CK	45.81 ± 0.41 ^c^	12.55 ± 0.78 ^b^	58.36 ± 1.07 ^d^
	N	73.91 ± 5.31 ^b^	13.18 ± 1.18 ^a^	87.09 ± 4.13 ^b^
	NG	153.91 ± 0.91 ^a^	13.89 ± 0.38 ^a^	167.80 ± 0.53 ^a^
	NM	39.28 ± 0.35 ^c^	4.50 ± 0.26 ^d^	43.78 ± 0.62 ^e^
	NMG	71.85 ± 2.40 ^b^	7.31 ± 0.28 ^c^	79.16 ± 2.12 ^c^
Total	CK	1364.63 ± 5.83 ^c^	430.05 ± 9.06 ^c^	1794.69 ± 14.89 ^c^
	N	1807.89 ± 14.82 ^b^	532.60 ± 9.16 ^b^	2340.49 ± 5.66 ^b^
	NG	2256.62 ± 20.69 ^a^	609.74 ± 0.24 ^a^	2866.36 ± 20.94 ^a^
	NM	1258.40 ± 92.51 ^c^	234.20 ± 4.34 ^d^	1492.59 ± 96.85 ^d^
	NMG	1749.54 ± 136.64 ^b^	465.81 ± 12.74 ^c^	2215.35 ± 149.38 ^b^

CK, N, NG, NM, and NMG are distilled water treatment, 40 mM NaCl treatment, 40 mM NaCl + 5 mM GABA treatment, 40 mM NaCl + 0.2 mM 3-MP treatment, and 40 mM NaCl + 0.2 mM 3-MP + 5 mM GABA treatment, respectively. Values in the same type and form of phenolic acid bearing different letters (^a,b,c,d,e^) are significantly different (*p* < 0.05).

**Table 3 antioxidants-10-00990-t003:** Effects of GABA on isoflavone content in soybean sprouts under NaCl stress (*n* = 3).

Isoflavone (µg/g DW)	Treatment				
	CK	N	NG	NM	NMG
Daidzin	481.44 ± 2.41 ^a^	430.85 ± 38.65 ^bc^	415.46 ± 2.39 ^c^	407.32 ± 11.47 ^c^	473.09 ± 11.17 ^ab^
Glycitin	27.33 ± 5.83 ^a^	26.72 ± 1.94 ^a^	33.32 ± 0.26 ^a^	29.50 ± 0.48 ^a^	31.48 ± 1.21 ^a^
Genistin	282.31 ± 5.54 ^bc^	284.67 ± 4.75 ^b^	294.21 ± 1.53 ^a^	265.90 ± 14.47 ^c^	311.22 ± 3.18 ^a^
Malonyldaidzin	2635.59 ± 17.69 ^a^	2620.03 ± 57.84 ^a^	2706.77 ± 25.32 ^a^	2207.63 ± 0.33 ^b^	2715.73 ± 148.84 ^a^
Malonylglycitin	322.59 ± 13.17 ^a^	268.01 ± 9.89 ^b^	262.00 ± 4.34 ^b^	231.27 ± 4.13 ^c^	273.04 ± 11.21 ^b^
Malonylgenistin	1681.32 ± 44.13 ^c^	2670.88 ± 44.91 ^b^	2860.55 ± 0.88 ^a^	2858.78 ± 48.41 ^a^	2717.28 ± 17.80 ^a^
Daidzein	61.35 ± 0.13 ^c^	105.01 ± 6.66 ^a^	78.15 ± 0.26 ^b^	77.56 ± 0.82 ^b^	111.98 ± 4.22 ^a^
Genistein	ND	40.92 ± 0.63 ^b^	38.10 ± 0.09 ^c^	36.81 ± 0.37 ^c^	42.65 ± 0.91 ^a^
Total	5436.92 ± 76.99 ^d^	6447.08 ± 161.30 ^b^	6689.56 ± 34.88 ^a^	6114.75 ± 17.30 ^c^	6676.46 ± 151.26 ^a^

ND meant not detected, CK, N, NG, NM, and NMG indicate distilled water treatment, 40 mM NaCl treatment, 40 mM NaCl + 5 mM GABA treatment, 40 mM NaCl + 0.2 mM 3-MP treatment, and 40 mM NaCl + 0.2 mM 3-MP + 5 mM GABA treatment, respectively. Values in the same line bearing different letters (^a,b,c,d^) are significantly different (*p* < 0.05).

## Data Availability

Not applicable.

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
