# Peer review of "GABA Regulates Phenolics Accumulation in Soybean Sprouts under NaCl Stress"

_antioxidants, 2021, doi:10.3390/antiox10060990_

Round 1
Reviewer 1 Report
Comments to the Authors of the manuscript antioxidants-1246545 "GABA regulates phenolics accumulation of soybean sprouts under NaCl stress".
Authors propose the report of an experiment aimed to investigate the role of GABA on phenolics accumulation and oxidative damage control in soybean sprouts under sodium chloride stress. The experiment was correctly designed and the manuscript well written. In my opinion, it represent a good contribution of knowledge to the field of study and may be suitable for pubblication after some minor changes.
1) First of all I suggest the Authors to improve the quality of the picture provided as Figure 1A. Please try to increase the contrast of the B/W picture.
2) Please also check that your results showed a significant increase of the phenylpropanoid compounds, which are precursors of the lignine biosynthesis. This increase is probably both linked tho the vascular tissue differentiation in the seedlings and to the tissue response to saline stress. In my opinion, you should improve your discussion with some considerations and citations on this physiological context.
Author Response
Comments to the Authors of the manuscript antioxidants-1246545 "GABA regulates phenolics accumulation of soybean sprouts under NaCl stress".
Authors propose the report of an experiment aimed to investigate the role of GABA on phenolics accumulation and oxidative damage control in soybean sprouts under sodium chloride stress. The experiment was correctly designed and the manuscript well written. In my opinion, it represent a good contribution of knowledge to the field of study and may be suitable for pubblication after some minor changes.
- First of all I suggest the Authors to improve the quality of the picture provided as Figure 1A. Please try to increase the contrast of the B/W picture.
Answer: The figure has been modified as suggested (revised version: line 428)
- Please also check that your results showed a significant increase of the phenylpropanoid compounds, which are precursors of the lignine biosynthesis. This increase is probably both linked to the vascular tissue differentiation in the seedlings and to the tissue response to saline stress. In my opinion, you should improve your discussion with some considerations and citations on this physiological context.
Answer: Thank you for your excellent suggestion. The purpose of this study was to investigate the of regulation of GABA on phenolics accumulation in soybean sprouts under NaCl stress. The mechanism of NaCl stress regulating the accumulation of phenolics substances in plants is not the purpose of this study. Moreover, we have not measured the difference of vascular tissue differentiation among each treatment and NaCl stress seemed to inhibit the growth of sprout (Figure 1A). Thanks again for your idea and we will investigate vascular tissue differentiation in our future study.
Reviewer 2 Report
Dear Authors,
I read the manuscript “GABA regulates phenolics accumulation of soybean sprouts under NaCl stress”
The research, even if not innovative, is well organized, data are clearly reported and results well justified. For all these reasons I consider the manuscript worth to be published in Antioxidants after minor revisions.
Here I report a few minor English inattention, together with some reference’s shortcoming, that I noticed.
Line 159: the 515 nm wavelength in the ABTS assay is incorrect.
Line190-91: The phrase “The content of free phenolics in soybean sprouts was about 20 times of bound phenolics” is not clear. May be it lacks something and it should be“20 times higher than that of bound phenolics”?
Line 263: “radical scavenging ability. And that showed” Delete the full stop.
Line 265 “which was 3~20 times of bound extract” change in “which was 3~20 times higher than that of bound extract”
Line 269: “the activity of these antioxidant enzymes. Indicating that” Delete the full stop
Line 293: “might due” change in “might be due”
Line 308: “This might be that GABA as a signal molecular mediated phenolics synthesis under NaCl stress” the phrase is not clearly written
Line 310: “increased” change with “increasing”
Line 337: “involved” change with “is involved”.
Line 347: cancel “editors” and put the editor of the book at the end of the reference.
Line 350: cancel “editor” and put the editor of the book at the end of the reference
Line 363: the name of the journal is lacking.
Best regards
Author Response
I read the manuscript “GABA regulates phenolics accumulation of soybean sprouts under NaCl stress”
The research, even if not innovative, is well organized, data are clearly reported and results well justified. For all these reasons I consider the manuscript worth to be published in Antioxidants after minor revisions.
Here I report a few minor English inattention, together with some reference’s shortcoming, that I noticed.
Line 159: the 515 nm wavelength in the ABTS assay is incorrect.
Answer: The wavelength in the ABTS assay has been modified in 734 nm (revised version: line 162)
Line190-91: The phrase “The content of free phenolics in soybean sprouts was about 20 times of bound phenolics” is not clear. May be it lacks something and it should be“20 times higher than that of bound phenolics”?
Answer: This sentence has been modified as suggested (revised version: line 189-190)
Line 263: “radical scavenging ability. And that showed” Delete the full stop.
Answer: The full stop has been deleted as suggested (revised version: line 244)
Line 265 “which was 3~20 times of bound extract” change in “which was 3~20 times higher than that of bound extract”
Answer: This sentence has been modified as suggested (revised version: line 246)
Line 269: “the activity of these antioxidant enzymes. Indicating that” Delete the full stop
Answer: The full stop has been deleted as suggested (revised version: line 250)
Line 293: “might due” change in “might be due”
Answer: This sentence has been modified as suggested (revised version: line 267)
Line 308: “This might be that GABA as a signal molecular mediated phenolics synthesis under NaCl stress” the phrase is not clearly written
Answer: This sentence has been modified (revised version: line 282)
Line 310: “increased” change with “increasing”
Answer: This sentence has been modified as suggested (revised version: line 284)
Line 337: “involved” change with “is involved”.
Answer: This sentence has been modified as suggested (revised version: line 310)
Line 347: cancel “editors” and put the editor of the book at the end of the reference.
Answer: This sentence has been modified as suggested (revised version: line 323-324)
Line 350: cancel “editor” and put the editor of the book at the end of the reference
Answer: This sentence has been modified as suggested (revised version: line 327-328)
Line 363: the name of the journal is lacking.
Answer: The name of the journal has been added (revised version: line 348)
Reviewer 3 Report
Comments to the paper: „GABA regulates phenolics accumulation of soybean sprouts
under NaCl stress”
I would like the authors to improve the following issues:
- Abstract – please structure abstract according to the scheme: introduction, methods, results and conclusions.
- |English correction is required. The article cannot be accepted in its current form.
- Introduction – the authors should broaden the information on the possible mechanisms of antioxidative GABA activity.
- The authors should develop the relevance of their paper. Now it is hardly possible to find the scientific signicance of this paper.
- Materials and methods section is described not sufficiently.
- Figures – the marking of statistical significance is completely unreadable. What „a/b/c” stand for? You should indicate which bars differer in comparison to other bars. The same with table 2, 3. How the values were compared.
- All methods should be validated. Provide coefficient of variability.
- How many experiments did you conduct? What is n in each experiment?
- Why the authors did not conduct antioxidant activity in cellular in vitro model?
- Did the authors have any positive control?
- Can the authors highlight the novelty of this paper?
Author Response
Comments and Suggestions for Authors Comments to the paper:
I would like the authors to improve the following issues:
- Abstract – please structure abstract according to the scheme: introduction, methods, results and conclusions.
Answer: The paragraph has been modified as suggested (revised version: line 11-24)
Abstract: NaCl stress causes oxidative stress in plants, which resist stress by synthesizing phenolics. The signaling molecule GABA can alleviate such abiotic stress, but the mechanism by which GABA works has yet to be studied. γ-aminobutyric acid (GABA) on phenolics accumulation in soybean sprouts under NaCl stress was investigated by measuring the influences of physiological biochemical, phenolic substance content, key enzyme activity, gene expression and antioxidant capacity of soybean sprouts. GABA reduced the oxidative damage of soybean sprouts caused by NaCl stress and enhanced total phenolics, phenolic acids and isoflavones content by 16.58%, 22.47%, and 3.75%, respectively. It also increased the activities and expression of phenylalanine ammonia lyase, cinnamic acid 4-hydroxylase, 4-coumarate coenzyme A ligase. Furthermore, GABA increased the antioxidant enzymes activities and antioxidant capacity as well. However, these events were inhibited by 3-mercaptopropionate (an inhibitor for GABA synthesis). Indicating that GABA mediated the phenolics accumulation and antioxidant system enhancement of soybean sprouts under NaCl stress.
- |English correction is required. The article cannot be accepted in its current form.
Answer: We have tried our best to improve the English of this manuscript.
- Introduction – the authors should broaden the information on the possible mechanisms of antioxidative GABA activity.
Answer: The paragraph has been modified as suggested(revised version: lines 41-49)
Past research has shown that GABA itself exhibits ROS scavenging ability [1]. Treatment of caragana roots with GABA under salt stress induced a variety of physiological and biochemical responses, such as signal transmission, protein degradation regulation, hormone synthesis, ROS formation, and polyamine metabolism [2], which indicates that GABA might be involved in the expression and regulation of salt-related genes in plants under salt stress. The content of GABA and phenolics in plants increased under salt stress [3] and exogenous GABA treatment [4]. Considering phenolics are not related to GABA metabolism, it can be speculated that GABA may mediate the synthesis of phenolics in soybean sprouts. However, the exact mechanism of its regulation on synthesis of phenolics is still unclear.
Reference:
[1] Liu, C., Li, Z., & Yu, G. . The Dominant Glutamic Acid Metabolic Flux to Produce γ-Amino Butyric Acid over Proline in Nicotiana tabacum Leaves under Water Stress Relates to its Significant Role in Antioxidant Activity. Journal of Integrative Plant Biology 2011, 53(8), 608–618.
[2] Shi, S.Q.; Shi, Z.; Jiang, Z.P.; Li-Wang, Q.I.; Sun, X.M.; Chun-Xiu, L.I.; Liu, J.F.; Xiao, W.F.; Zhang, S.G. Effects of exogenous GABA on gene expression of Caragana intermedia roots under NaCl stress: regulatory roles for H2O2 and ethylene production. Plant Cell & Environment 2010, 33, 149-162.
[3] Ma, Y.; Wang, P.; Chen, Z.; Gu, Z.; Yang, R. NaCl stress on physio-biochemical metabolism and antioxidant capacity in germinated hulless barley (Hordeum vulgare L.). Journal of the Science of Food and Agriculture 2019, 99, 1755-1764.
[4] Ma, Y.; Wang, P.; Wang, M.; Sun, M.; Gu, Z.; Yang, R. GABA mediates phenolic compounds accumulation and the antioxidant system enhancement in germinated hulless barley under NaCl stress. Food Chemistry 2019, 270, 593-601.
- The authors should develop the relevance of their paper. Now it is hardly possible to find the scientific signicance of this paper.
Answer: The scientific significance of our research is to clarify the mechanism of GABA as a signal molecule in regulating the synthesis of phenolics in soybean sprouts under NaCl stress: GABA regulated the phenolics synthetase activity and gene expression in soybean sprouts to promote the enrichment of phenolics, thereby enhancing the antioxidant capacity and stress tolerance, and strengthening the stress defense of soybean sprouts und NaCl stress.
- Materials and methods section is described not sufficiently.
Answer: We have modified the description of the materials and methods section as suggested (revised version: line 146-148)
- Figures – the marking of statistical significance is completely unreadable. What „a/b/c” stand for? You should indicate which bars differer in comparison to other bars. The same with table 2, 3. How the values were compared.
Answer: The description of the marking of statistical significance has been added (revised version: line 411-413, 418-420, 425-427, 447-450, 456-458)
- All methods should be validated. Provide coefficient of variability.
Answer: The measurement of each index refers to the methods in the literatures. These methods have been proved to be reliable by other researchers.
- How many experiments did you conduct? What is n in each experiment?
Answer: We conduct three biological replicates in each experiment, and we have added n=3 in the corresponding table or figure (revised version: line 413, 420, 427, 450, 458)
- Why the authors did not conduct antioxidant activity in cellular in vitro model?
Answer: The purpose of this study was to investigate the mechanism of GABA regulating phenolics accumulation of soybean sprouts under NaCl stress. We measured the activities of CAT, POD and SOD and free radical scavenging ability to prove that GABA treatment enhanced the antioxidant enzymes activity of soybean sprouts and improved the ABTS and DPPH free radical scavenging abilities. According to our knowledge, results of ABTS and DPPH free radical scavenging were consistent with results in cellular in vitro model. Therefore, we did not conduct antioxidant activity in cellular in vitro model.
- Did the authors have any positive control?
Answer: We only investigated the antioxidant capacity of sprouts extracts. The negative control was used, but positive controls like VC etc. were not used since the manuscript mainly focused on the mechanism of GABA on phenolics accumulation under NaCl stress, and the antioxidant capacity was only used to identify the relationship between phenolics content and antioxidant capacity.
- Can the authors highlight the novelty of this paper?
Answer: The novelty of this paper has been highlighted (revised version: line220-230, 315-317). Effects of GABA on isoflavone content in soybean sprouts under NaCl stress showed that GABA treatment promoted the acetylation and malonylation of isoflavone (table 3), which might be crucial for the increase of free radical scavenging ability of soybean sprouts.
Round 2
Reviewer 3 Report
Unfortunately, the authors did not correct the manuscript according to my suggestions.
The statistic is still not clear since the authors did not indicate what a, b or c mean. Now I get the impression that everything is different from everything else. You have to indicate that a means significant difference between samples treated with the other one. For instance, figure 2, CK column - there is a letter d - what does it mean? Please indicate clearly with which other samples it is statistically different.
Now the manuscript cannot be accepted.
Author Response
We have modified all the legends and Fig 1 as well as Fig 3. We hope these will meet reviewer’s requirement.
Values in the same measured category bearing different letters are significantly different (p <0.05). For instance, in Fig 2 B, CK and NMG were not significantly different because both of them bearing letter c, but they were significantly different with others. In the figures with two measured category (i.e. Fig1 B, C, D; Fig3 A, B, D), lowercase and uppercase letters were used in the present manuscript to avoid misleading.
Round 3
Reviewer 3 Report
I don't have any more comments, but the paper is of low quality.